# Vessel-on-a-Chip: A Powerful Tool for Investigating Endothelial COVID-19 Fingerprints

**DOI:** 10.3390/cells12091297

**Published:** 2023-05-02

**Authors:** Oksana Shevchuk, Svitlana Palii, Anastasiia Pak, Nuria Chantada, Nuria Seoane, Mykhaylo Korda, Manuel Campos-Toimil, Ezequiel Álvarez

**Affiliations:** 1Department of Pharmacology and Clinical Pharmacology, I. Horbachevsky Ternopil National Medical University, 46001 Ternopil, Ukraine; shevchukoo@tdmu.edu.ua (O.S.); palij_svmy@tdmu.edu.ua (S.P.); 2Department of Medical Biochemistry, I. Horbachevsky Ternopil National Medical University, 46001 Ternopil, Ukraine; pak_aniv@tdmu.edu.ua (A.P.); korda@tdmu.edu.ua (M.K.); 3Departamento de Farmacología, Farmacia y Tecnología Farmacéutica, Universidade de Santiago de Compostela, 15782 Santiago de Compostela, Spain; antuvio08@gmail.com (N.C.); ezequiel.alvarez.castro@gmail.com (E.Á.); 4Physiology and Pharmacology of Chronic Diseases (FIFAEC) Center for Research in Molecular Medicine and Chronic Diseases (CiMUS), University of Santiago de Compostela, 15782 Santiago de Compostela, Spain; nuria.seoane@rai.usc.es; 5Instituto de Investigación Sanitaria de Santiago de Compostela (IDIS), Complexo Hospitalario Universitario de Santiago de Compostela (CHUS), SERGAS, Travesía da Choupana s/n, 15706 Santiago de Compostela, Spain; 6CIBERCV, Institute of Health Carlos III, 28220 Madrid, Spain

**Keywords:** COVID-19, endothelial dysfunction, long COVID, microfluidic system, personalized COVID-19 follow-up, vessel-on-a-chip model

## Abstract

Coronavirus disease (COVID-19) causes various vascular and blood-related reactions, including exacerbated responses. The role of endothelial cells in this acute response is remarkable and may remain important beyond the acute phase. As we move into a post-COVID-19 era (where most people have been or will be infected by the SARS-CoV-2 virus), it is crucial to define the vascular consequences of COVID-19, including the long-term effects on the cardiovascular system. Research is needed to determine whether chronic endothelial dysfunction following COVID-19 could lead to an increased risk of cardiovascular and thrombotic events. Endothelial dysfunction could also serve as a diagnostic and therapeutic target for post-COVID-19. This review covers these topics and examines the potential of emerging vessel-on-a-chip technology to address these needs. Vessel-on-a-chip would allow for the study of COVID-19 pathophysiology in endothelial cells, including the analysis of SARS-CoV-2 interactions with endothelial function, leukocyte recruitment, and platelet activation. “Personalization” could be implemented in the models through induced pluripotent stem cells, patient-specific characteristics, or genetic modified cells. Adaptation for massive testing under standardized protocols is now possible, so the chips could be incorporated for the personalized follow-up of the disease or its sequalae (long COVID) and for the research of new drugs against COVID-19.

## 1. Introduction

The global spread of the novel severe acute respiratory syndrome coronavirus 2 (SARS-CoV-2) triggered a pandemic that began in December 2019, leading to over 677 million reported cases worldwide (https://www.worldometers.info/coronavirus/, accessed on 10 February 2023). As of February 2023, the World Health Organization (WHO) states that the COVID-19 pandemic is at an “inflexion point” where high levels of immunity to the virus (due to the natural course of disease and/or vaccination) are starting to limit its impact and reach. Nevertheless, SARS-CoV-2 continues to cause a large number of new infections around the world, and while the Omicron variant with subvariants appears to be associated with lower severity than Delta, the severity is still influenced by individual factors such as age, comorbidity, and vaccination status [1]. Moreover, the emergence of new variants such as the Kraken subvariant of the virus [2] means that the question of when COVID-19 will cease to be a global emergency remains unanswered.

Although essentially a pneumological virus, SARS-CoV-2 primarily affects the respiratory system; patients with comorbidities such as acute kidney disease, chronic obstructive pulmonary disease, arterial hypertension, cardiac diseases, diabetes, obesity, or cancer are at the highest risk of mortality. Therefore, this knowledge should be used for future research, control and prevention of COVID-19 [3]. In addition, COVID-19 has significant effects on the cardiovascular system. The vascular response to the “cytokine storm” produced by the infection and the interaction between SARS-CoV-2 and angiotensin-converting enzyme 2 (ACE2) can result in a substantial reduction in cardiac contractility and subsequent myocardial dysfunction. Furthermore, many patients do not fully recover vascular function and continue to experience symptoms even in the absence of detectable viral infection [4].

A significant proportion of COVID-19 patients have not fully recovered and continue to experience lingering symptoms for months or even years after the initial infection. This novel clinical syndrome has been referred to as “long COVID”, “post-acute sequelae of COVID-19”, or “post-COVID-19 conditions” [5,6,7,8].

These post-acute sequelae of COVID-19 cause cardiovascular injury, particularly in the vascular endothelium (Positioning of European Society of Cardiology about the endothelium in COVID [9]). Therefore, in this post-COVID-19 era, it is crucial to monitor and study vascular function with the best available resources and technology. In this review, we will analyze the current knowledge on the main vascular sequelae of COVID-19 and, as the disease continues, explore the potential benefits of using new microfluidics chip technology to increase our understanding of the underlying mechanisms and to integrate clinical findings through advanced and personalized monitoring of vascular function. To date, 3D printers and microfluidics have been used to combat the COVID-19 pandemic by providing personal protective equipment (masks, respirators, face shields, goggles, and isolation chambers/hoods) to diagnostic supplies (sampling swabs and lab-on-a-chip) and supportive care (respiratory equipment) [10]. However, it has been suggested that their contribution should be greater, to support new models for viral infections that include the multicellular interactions that the 2D culture models negate, and avoid the problems of the interspecies differences of the animal models [11], accelerating the diagnosis and monitoring of the disease.

## 2. Vascular Fingerprint of COVID-19

### 2.1. SARS-CoV-2 Gate

SARS-CoV-2 belongs to the family of coronaviruses known as human coronaviruses (HCoVs), which also includes the Middle East respiratory syndrome coronavirus (MERS-CoV), which is the cause of an epidemic that occurred in June 2012 [12,13,14]. Most commonly, seasonal respiratory diseases such as the common cold, bronchiolitis, and pneumonia are caused by a group of HCoVs [15].

SARS-CoV-2 is a Betacoronavirus, with a positive-sense, single-stranded RNA genome and an envelope. Bats are believed to be the natural host for SARS-CoV-2, as they share a genome with other SARS-like coronaviruses, such as bat-SL-CoVZX45 and bat-SL-CoVZX2. The virus is mainly transmitted through close person-to-person contact [16].

ACE2 has been identified as the main entry receptor for SARS-CoV-2. This receptor is widely distributed in many tissues, including the endothelial cells of the arteries, arterioles, and venules of the heart and kidney. ACE2 plays an important role in regulating blood pressure, neural, pulmonary, renal, cardiovascular, and immune homeostasis through the renin–angiotensin system (RAS) in various organs such as the lungs, heart, brain, kidney, blood vessels, gastrointestinal tract, and testes [17,18,19,20].

ACE2 is a crucial player in RAS. The RAS may function in a two-axis-molecular cascade way, with the classical axis composed of ACE, angiotensin II (Ang II), and the angiotensin type I receptor (AT1R) [21,22]. This ACE–Ang II–AT1 pathway is responsible for vasoconstriction, fibrosis, inflammation, cellular growth, migration, cardiac hypertrophy, thrombosis, and reactive oxygen species (ROS) production [23]. In contrast, the counter-regulatory axis is formed by ACE2, Ang-(1–7), and the Mas receptor (MasR) [24]. Ang-(1–7) has properties such as anti-inflammatory, antifibrotic, antiproliferative, antioxidative, vasodilator, and antithrombotic, which are mainly exerted via the MasR [21]. The ACE2 primary function is to metabolize Ang II to Ang 1–7, reducing inflammation and oxidative damage, and counteracting the harmful effects of the ACE–Ang II–ATR1 pathway [23,25,26,27] (Figure 1). Ang II can bind to two types of receptors, AT1R, and AT2R. The former leads to vasoconstriction, fibrotic remodeling, inflammation, and enhanced ROS production, while the latter is responsible for vasoconstriction and growth inhibition [21,28].

Ang II is a pro-oxidant peptide that influences the activation of nicotinamide adenine dinucleotide phosphate (NADPH) oxidases in various vascular cell types, including endothelial cells. This results in the uncontrolled formation of superoxide anions, which cannot all be converted into H_2_O_2_ by superoxide dismutase. As a result, a significant amount of peroxide anions reacts with nitric oxide to form peroxynitrite, which further damages blood vessels [28]. This causes further oxidative damage to various biomolecules, such as proteins, lipids, and DNA [29,30]. An imbalance between ACE and ACE2 can lead to excessive Ang II formation, which can contribute to endothelial cell damage and further exacerbate the oxidative stress [31].

ACE2 is known to have a range of biological functions in different diseases, including heart failure, myocardial infarction, hypertension, kidney diseases, acute lung injury, and diabetes. Some studies suggest that genetic determinants that downregulate ACE2 protein expression and subsequently dysregulate the RAS are associated with the severity of COVID-19 [32]. In addition to ACE2, other host entry factors and alternative receptors for SARS-CoV-2, such as C-type lectins, DC-SIGN and L-SIGN (dendritic cell-specific intercellular adhesion molecules (ICAM)-3 grabbing non-integrin and liver/lymph node-specific ICAM-3 grabbing non-integrin), have also been proposed [33].

SARS-CoV-2 primarily affects lung tissues, leading to the development of interstitial pneumonitis and acute respiratory distress syndrome (ARDS). However, the virus has also been shown to affect the cardiovascular system [34,35]. The first weeks and months of the pandemic demonstrated that COVID-19 is not solely a respiratory illness, as it is associated with vascular thrombosis events and affects endothelial cells [31,36,37,38]. Importantly, ACE2 is also expressed in endothelial cells [39].

Various clinical studies have identified age and comorbidities with substantial endothelial dysfunction, such as hypertension, obesity, diabetes mellitus, and coronary heart disease, as risk factors for a more severe COVID-19 course, “cytokine storm”, ARDS development, oxygen therapy (invasive/non-invasive) need, and higher mortality rates [25,40,41,42,43,44,45,46]. These conditions are known to be associated with low-grade inflammation, oxidative stress, and cytokine production, which increase the risk of thromboembolic complications and vital organ damage [8,47,48,49,50,51]. Inflammation-induced endotheliitis, alterations in blood flow and viscosity, and neutrophil extracellular trap formation have been linked to microvascular thrombosis, which is crucial to the course and severity of COVID-19 [52]. Therefore, inflammation, endothelial dysfunction, hypercoagulation, hypofibrinolysis, and RAS alteration are the central players in the pathophysiology of COVID-19.

### 2.2. ACE2 and COVID-19 Severity and Medical Interventions

During the early months of the COVID-19 pandemic, there was uncertainty about the safety of taking ACE-inhibitors (ACEI) and angiotensin receptor blockers (ARB). These drug classes are commonly used as first-line treatments for hypertension, congestive heart failure, and other cardiovascular diseases. High blood pressure and diabetes mellitus are the most common comorbidities associated with severe COVID-19 [53,54]. Experimental data showed that ACEI (captopril, enalapril, perindopril, and other -prils) as well as ARB (losartan, olmesartan, telmisartan, etc.) could increase the levels of ACE2, which is possibly due to a decrease in the enzyme substrate—angiotensin II [55].

The logic was next—the use of ACEI and/or ARB could increase susceptibility to COVID-19 infection by increasing the levels of ACE2, which is the entry point for the host target cells. This led to the idea of blocking the entry of the coronavirus via virus- or ACE2-directed strategies. The randomized clinical trial “ACE2 Chewing Gum on SARS-CoV-2 Viral Load (COVID-19)” is still ongoing (https://clinicaltrials.gov/show/NCT05433181, accessed on 10 February 2023). Some promising molecules (monoclonal antibodies) were designed as antiviral agents [56].

The age is a well-established risk factor in determining the severity and outcomes of COVID-19. However, there is conflicting data on the expression of ACE2 across different age groups. While some studies have reported lower levels of ACE2 protein in children compared to adults, others have reported the opposite [57].

It is worth noting that the impact of ACEI/ARB on ACE2 expression and COVID-19 severity is still a topic of debate, with conflicting findings reported across studies. For instance, Lee et al. conducted a detailed study that showed no influence of these drugs on the expression of ACE2 in respiratory cilia, which is the main area for infection [58]. Nevertheless, some studies suggest that ACEI/ARB may reduce systemic inflammation by lowering angiotensin II levels. Notably, higher levels of plasma soluble ACE2 have been reported in older individuals and men compared to women [59], as well as in comorbidities such as metabolic syndrome, obesity, and hypertension, which are associated with severe COVID-19 [60]. Conversely, other studies have proposed an inverse relationship between ACE2 expression and disease severity, suggesting that the main functional role of ACE2 may be in reversing the inflammation process [57]. Thus, further comprehensive studies are needed to elucidate the true relationship between ACEI/ARB and COVID-19 severity, and organ-on-chip techniques could serve as a useful alternative to investigate the underlying molecular mechanisms.

### 2.3. Endothelial Dysfunction and COVID-19

The vascular endothelium is a monolayer that separates blood flow from tissues, lining the inner walls of blood vessels with an approximate area of 7000 m^2^ [61]. Endothelial cells play a crucial role in angiogenesis through the actions of vascular endothelial growth factor and its receptor, angiopoietin 2, hypoxia-inducible factor activation, glutamine and asparagine metabolism, von Willebrand factor (vWF), and the jagged 1-NOTCH1 (neurogenic locus notch homolog protein 1) signaling pathway [62]. Endothelial dysfunction is defined as the inability of the endothelium to maintain vascular homeostasis.

The vascular endothelium plays a crucial role in maintaining hemostasis by serving as the interface between the bloodstream and tissues. It contributes to coagulation via the glycocalyx and by synthesizing procoagulant factors such as vWF, tissue factor, thromboxane A2 (TXA2), and plasminogen activator inhibitor 1 (PAI1). Additionally, it synthesizes anticoagulant factors including thrombomodulin, tissue plasminogen activator (tPA), tissue factor pathway inhibitor, and prostaglandin I2 (PGI2) [63].

Another function of the vascular endothelium is to regulate vascular tone through the release of both vasoconstrictors, such as endothelin-1, TXA2, and prostaglandin H2, and vasodilators, such as nitric oxide, hydrogen sulfide (H_2_S), endothelium-derived hyperpolarizing factor, and PGI2 [38,51,64]. Endothelial cells also play a role in inflammation by synthesizing factors such as ICAM-1, vascular cell adhesion molecule 1, E-selectin, and monocyte chemoattractant protein-1 (MCP1). Interestingly, the hormone leptin, which is an important factor in obesity, promotes a decrease in nitric oxide and an increase in MCP1, leading to vasoconstriction and high leukocyte infiltration in the vessel wall [65,66,67].

Several facts suggest the direct infection of endothelial cells via ACE2 by SARS-CoV-2. The virus has a 10 times higher affinity to ACE2 compared to other SARS-CoV viruses [68]. Post-mortem examinations have revealed the presence of viral inclusion structures, an accumulation of inflammatory cells associated with the endothelium, and apoptotic bodies in the heart, small bowel, and lung tissues [51]. These findings suggest that SARS-CoV-2 induces vasculitis, leading to microcirculatory alterations and subsequent clinical complications. In Italy and the USA, cases of Kawasaki disease were found in children following confirmed COVID-19 [69]. These findings provide further evidence that inflammation, damage, and dysfunction of the endothelium play a central role in the pathogenesis and pathophysiology of COVID-19, contributing to disease severity.

The COVID-19-induced cytokines hyperproduction (“cytokine storm”) also disrupts endothelial function. The excessive production of pro-inflammatory cytokines (such as interferon-α, and -γ, interleukin (IL)-1β, IL-6, IL-12, IL-18, IL-33, tumor necrosis factor α, and tumor growth factor β), colony-stimulating factors, and chemokines (such as C-X-C motif chemokine ligand (CXCL) 10, CXCL8, CXCL9, C-C motif chemokine ligand (CCL) 2, CCL3, and CCL5) sustains the systemic inflammatory response [26,70,71,72]. The efflux of cytokines promotes microvascular disorders, increases vascular permeability, and leads to alveolar edema and hypoxia [70]. It also promotes procoagulative and proadhesive changes and hypercoagulation [73].

Randomized controlled trials (RCTs) evaluating IL-1 blockers (anakinra, canakinumab) for the treatment of patients with COVID-19 did not find evidence for their significant beneficial effects [74]. On the other hand, the efficacy of IL-6 blocking agents (tocilizumab, sarilumab) in the treatment of COVID-19 showed that tocilizumab reduces all-cause mortality compared to standard care alone or placebo, but the increase in clinical improvement outcome was limited or absent [75].

One characteristic of endothelial dysfunction and thrombotic events is the inhibition of nitric oxide-synthase resulting in nitric oxide deficiency [37]. Changes in nitric oxide bioavailability, oxidative stress activation, and inflammation contribute to the inability of endothelial function [76,77].

A healthy vascular endothelium supports the normal functioning of the blood coagulation system, regulation of vasodilation, fibrinolysis, and the balance of thrombosis. This is possible because endothelial cells present a dynamic system and synthesize a large number of tissue factors, prostacyclin, thrombomodulin, PAI1, tPA, plasminogen, nitric oxide, and protein C activators, among others [78]. Endothelial dysfunction leads to the depletion of protective endogenous factors, while the concentration of endothelin-1, angiotensin II, vWF, and PAI1 rises [79]. In addition, endothelial dysfunction is associated with the predominant influence of vasoconstrictive, pro-thrombotic, and pro-inflammatory factors [80].

Endothelin-1 is a highly significant vasoconstrictor, consisting of 21 amino acids, with two other endothelin isopeptides—endothelin-2 and endothelin-3. All isoforms are released not only from endothelial cells but also from cardiac myocytes, vascular smooth muscle cells, renal tubular epithelial cells, glomerular mesangial cells, glial cells, pituitary cells, macrophages, mast cells, and other cells [81].

### 2.4. Long COVID-19

According to clinicaltrials.gov, there are 8718 search results for COVID-19 RCTs: 3345 are completed, 389 were terminated, 52 were suspended, and 1155 have an unknown status. For post-COVID-19 conditions, there are 277 RCTs in the entire database, with 69 completed, 123 currently recruiting or enrolling participants by invitation, and 40 studies registered but not recruiting at present.

The post-infection syndrome is not unique to COVID-19 and can occur with both viral (e.g., influenza, Ebola, mononucleosis) and non-viral infections (Lyme disease, giardiasis, etc.). WHO defines “long COVID” as the condition where symptoms (such as fatigue, shortness of breath, chest pain, cognitive dysfunction, palpitation, and others) persist for at least 3 months after probable or confirmed SARS-CoV-2 infection with a minimum duration of 2 months and cannot be explained by an alternative diagnosis [82]. Between 6 and 10% of patients after acute COVID-19 may experience long-term symptoms that impair their daily activities [4,83,84], representing a significant number of individuals. Long COVID can impact various organs and systems, causing a wide range of pathologies and symptoms, from neurological symptoms such as brain fog to chronic fatigue syndrome, postural orthostatic tachycardia syndrome, diabetes, erectile dysfunction, and many others [4,5,6,84,85,86,87]. Risk factors for long COVID include low immune response with low anti-SARS-CoV-2 IgG levels, infection with the Delta virus variant, mild form of disease, female sex, Hispanic or Latino ethnicity, type 2 diabetes, the presence of specific autoantibodies, connective tissue disorders, and pre-existing allergic conditions, among others [4]. However, it is important to note that approximately one-third of people with long COVID have no identified pre-existing conditions. There are suggestions that the risk of long COVID is lower in vaccinated individuals [88].

Endothelial dysfunction plays a critical role in the severity of acute COVID-19 and continues to be a factor in the post-COVID-19 period. There are theories suggesting that children’s uncompromised endothelial systems may protect them from the most severe complications of COVID-19 [61,66,89].

Endothelium-dependent flow-mediated dilation (FMD) is a non-invasive endothelial function test (a vascular function test) and an established method of evaluating future cardiovascular disease risk [80]. A study conducted in the Ternopil region (Ukraine) analyzed post-COVID-19 patients up to 3 months after their last negative polymerase chain reaction test. The study revealed that 100% of critical patients who survived and 90.2% of oxygen-dependent patients with severe COVID-19 had vasoconstriction during the FMD test, while 15.9% and 11.83% of patients with moderate and mild courses of the disease, respectively, showed such reactions. These findings suggest that the severity of SARS-CoV-2 infection has a significant impact on endothelial function deterioration [90].

One mechanism that is thought to be responsible for COVID-19-related cardiovascular injury is the expression of ACE2. The high expression of this enzyme in vessels and endothelium explains the broad tissue tropism of SARS-CoV-2, the multisystem injury, hypercoagulation, and microvascular thrombosis seen in COVID-19, as well as the worse prognosis in cases of comorbidity with endothelial dysfunction [33,45,91,92]. In addition, systemic inflammation and increased inflammatory signaling also play a role [78,85,93,94].

The studies conducted on long COVID patients are highly heterogeneous, but they suggest that this is a multisystem disease with a wide spectrum of clinical symptoms. The main mechanisms underlying its pathophysiology are not well defined, but they are believed to be connected to specific long-lasting inflammation and persisting endothelial dysfunction. A deeper understanding of long COVID is necessary, and further studies are needed to shed light on this complex and poorly understood condition.

Altogether, these bring us to some of the position statements of the European Society of Cardiology [9]. In studies of COVID-19 outcomes and treatments, it is important to monitor the biomarkers and function of endothelial cells, such as FMD and arterial stiffness. High-quality data collection is necessary, including follow-up studies of survivors, as there is a lack of information on endothelial cells function testing in COVID-19. Essentially, the idea is that monitoring the health of the endothelial cells and collecting good quality data on their function is important in COVID-19 research, and more research needs to be conducted in this area.

The effects of SARS-CoV-2 on cardiovascular health caused by virus-mediated endocytosis and downregulation of ACE2 are still not fully understood. However, ongoing clinical trials utilizing recombinant ACE2 may offer valuable insights. Finally, it is important to study the long-term cardiovascular effects in patients after they have recovered from COVID-19 in order to implement preventative measures in a timely manner. One potential method of detecting early vascular complications post-COVID-19 is to measure endothelial function as well as myocardial injury and respiratory function markers in convalescent patients.

## 3. Contribution of Microfluidic Technology

Can microfluidic in vitro models, such as vessel-on-a-chip systems, be employed to investigate the pathophysiology of COVID-19 in endothelial cells, develop personalized patient follow-up strategies post-COVID-19, or develop novel drugs to address vascular complications of COVID-19?

The critical points to consider in this issue include [9]:Determining the main effects of SARS-CoV-2 on endothelial function, including activation of endothelial cells, recruitment of leukocytes, and platelet activation. Factors such as cellular senescence, oxidative stress, and other aging-related features should also be taken into consideration, along with gender influence. Therefore, microfluidic chips should be personalized to address these concerns.Exploring the effects of common cardiovascular drugs on endothelial responses to SARS-CoV-2.

These points can be considered and miniaturized using microfluidic technology in vessel-on-a-chip systems specifically designed for COVID-19 research. However, with regard to this, some reflection should be undertaken to answer the question of whether these new in vitro models can be a valuable tool for vascular research in the COVID-19 era. We have focused our analysis on the following topics:Contributions of miniaturized biomimetic endothelium in vitro models.Pharmacological relevance of endothelial functions in microfluidic chips.Advantages of vessel-on-a-chip systems with respect to 2D classical in vitro models.“Personalization” of endothelial models.Tools for “point-of-care” solutions, diagnosis, or follow-up.

### 3.1. Contributions of Miniaturized Biomimetic Endothelium In Vitro Models

Microfluidic models offer several advantages, including the incorporation of a 3D structure that mimics the physiological situation, allowing for the spatial distribution and orientation of cells. Additionally, microfluidic models can incorporate flow across the system, addressing a key issue in the study of vascular tissue. However, although the first commercial microfluidic chips provide a wide range of structures and channel geometries, they suffer from two important issues. Firstly, the internal lumen of the channels has a squared-shape cross-section with 90° angles in their corners, which is not biomimetic. Secondly, the materials used, unless biocompatible, are not biological (polystyrene, polyethylene, or silicones), which are normally hydrophobic and do not provide a biological environment. Researchers have proposed solutions to these issues, and various approaches have been suggested.

More realistic round sections of vessels’ inner walls have been achieved using various fabrication techniques that can be categorized into two groups: microfluidic-based vascular models and self-assembled vascular networks within cell-laden matrices [95]. The first category includes the casting–peeling–bonding scheme [96], templating scheme [97], and 3D microfluidic bioprinting [98], which can even incorporate a two-photon polymerization laser technique for high-definition bioprinting [99]. A review of these techniques can be found in [100]. All these approaches allow for the creation of biomimetic 3D configurations of the vascular network and the reproduction of structural key issues of vessels’ bifurcations or stenosis.

The problem of creating a biological environment surrounding the vessels of the chip has been addressed in two ways. The first approach, which was mentioned earlier, is 3D microfluidic bioprinting. This approach involves using biological matrixes or biomimetic hydrogels as the “ink” to print the vessels on the chip. While this method allows for the on-demand design of the vascular network, it requires a 3D printer that may not be available in all laboratories. The second approach, which has great potential, involves studying new biologic materials for the construction of the extracellular matrix (ECM) inside the chips. Initial vessel-on-a-chip models consisted of a channel made of plastic, glass, or silicone material, in which the internal walls were cultured with endothelial cells. The next step is to reconstitute the “sub-endothelial space” by including an appropriate ECM in the chip.

Gelatin, a hydrolysate of collagen [101], collagen [102,103,104], fibrinogen [105], fibrin [106], gelatin methacrylate hydrogel [107], gelatin methacryloyl [108], calcium–alginate hydrogel [109], and a mix of different hydrogels [99,110], among others, are some of the most commonly used materials to recreate the ECM environment.

Another approach involves the use of a membrane composed of a polymeric blend of poly-caprolactone/chitosan to seed human endothelial cells, with the goal of enabling resistance to the trans-endothelial hydraulic pressure of flow. Tuning the membrane with different percentages of the two components allows for the optimization of hydraulic resistance by enhancing cellular adhesion [111]. Each of these proposals has its advantages and disadvantages, but all aim to recapitulate key aspects of the subendothelial space in vascular tissues. Using these different approaches, microfluidic chips have been developed with a monolayer of endothelial cells seeded into the lumen of the channels, which demonstrate good confluence and can be maintained in the incubator for several days [112].

It is important at this stage to maintain the barrier function of the endothelial monolayer seeded on the ECM, which should control the diffusion of molecules from the lumen of the “vessel” to the “subendothelial” space that is now filled with the ECM. The barrier function in the presence of an endothelial monolayer can be easily tested in these chips by infusing a fluorescent probe with a controlled molecular weight (typically 40 kDa) [105,107,110].

The properties of this kind of ECM also include the ability to create chemokine or growth factor gradients in this space [101,102,103,113], which can be used to analyze endothelial functions such as neoangiogenesis [102,104,114] or the self-assembled vascular network [115,116], as we will discuss below.

The oxygen tension [109] in the ECM surrounding the endothelialized engineered vessel is also an important factor to consider when seeding different types of cells in the “subendothelium”, such as pericytes [104,105], mesenchymal progenitor cells [99,107], and smooth muscle cells [108,109]. This property is also crucial for mimicking the irrigation function of the capillary network [116,117].

Last but not least, the ECM where the endothelial cells are seeded is important to maintain their physiological phenotype [108,110,115]. This is important for the pharmacological relevance of the microfluidic chip created, as we will see in the following sections.

The incorporation of a bioactive ECM, in addition to its contribution to other properties, lays the groundwork for an important challenge in organ-on-a-chip technology and tissue engineering: the co-culture of relevant cell types in a 3D biomimetic structure. Vascular tissue is composed of several layers, with each layer being predominantly occupied by a specific cell type. Typically, from the lumen, the layers of a vessel are the endothelium, consisting of a monolayer of endothelial cells, a basal protein lamina, the lamina media, composed of pericytes or several layers of smooth muscle cells, and adventitia, where fibroblasts predominate, but which is infiltrated by vasa vasorum, nervous endings, and immunity cells. The efforts to reproduce this in microfluidic chips are simplified into two main approaches. On the one hand, tissue cells are embedded in the ECM surrounding the endothelial monolayer, with pericytes [104,105], mesenchymal progenitor cells [99,107], and smooth muscle cells [108,109] being the main examples. On the other hand, a complete muscular layer is created trying to mimic medium or large vessels [109,118], or even the myocardial muscle [119].

All these approaches and efforts have been aimed at creating microfluidic chips that can replicate the structural and functional properties of biomimetic channels, which can support the flow of solutions through their lumens. To achieve this, the system must be able to withstand the pressure of the flowing liquid and the cells seeded within it. In particular, the endothelial cells in the lumen should support the shear stress and remain stable during the experiments. In other words, microfluidic systems provide the opportunity to deeply analyze the effects of shear stress [120,121] and transmural pressure [122], as well as the influence of different flow patterns such as pulsatile or continuous flow or cyclic stretch [111,112] (Figure 2).

### 3.2. Pharmacological Relevance of Endothelial Functions in Microfluidic Chips

In addition to the biomimetic structural organization and environment discussed in the previous point, vessel-on-a-chip models must also provide pharmacological relevance, meaning that the experiments conducted in these models should allow for conclusions that could be rationally extrapolated to animals or humans. To achieve pharmacological relevance, microfluidic models must closely mimic the physiological biomimetics of vascular tissue. This requires that cells in the chip exhibit normal vascular cell functions and respond physiologically to stimuli, such as deleterious factors or drugs, within the model.

The most explored functions in vessel-on-a-chip models have been the barrier function, neoangiogenesis, formation and perfusion of capillary networks, and various parameters of endothelial cell activation and interaction with circulating cells [112].

#### 3.2.1. Barrier Function

As mentioned previously, one of the first functions to be modeled in a microfluidic vessel is the barrier function. In normal vessels, the architecture presents resistance to the diffusion of large molecules and the transmigration of cells. To test the barrier function, the permeability coefficient is typically quantified as the number of molecules of a certain size that can pass through the endothelial monolayers, using fluorescent probes of known molecular weight [105,107,110,123]. Pathophysiological processes, such as inflammation, infection, apoptosis, or necrosis, can alter the barrier function, making the endothelium more permeable. Additionally, abnormal or anarchic architecture, as seen in cancer tumors, can significantly affect the function of the barrier. The improper organization of vascular growth can also dramatically alter its function [124,125].

Good models of vessel-on-a-chip demonstrating a measurable barrier function have been developed using only one type of cell, typically endothelial cells [110], although in some cases, the strength of the barrier has been compared with models that incorporate other types of cells [126]. Interestingly, in these models, electrochemical methods have been developed to measure the endothelial barrier function using electrical impedance. This allows for independence from optical (microscopic imaging) methods and likely accelerates real-time data acquisition [126,127].

Other systems enhance their complexity by introducing additional types of cells in the ECM besides the endothelial cells that form the luminal barrier [105,107]. These miniaturized systems have demonstrated the ability to respond in a physiological manner to drugs or stimuli that affect the barrier function, thus providing valuable insights into vascular physiological properties in an in vitro system. For instance, these chips have shown responses to thrombin [105], histamine, or tumor necrosis factor-α [128] as well as to chemicals such as ethylenediaminetetraacetic acid or shear stress [126].

The blood–brain barrier (BBB) is one of the most complex systems to model on a chip. This barrier is crucial for transporting drugs into the central nervous system, but simulating it in an organ-on-a-chip requires the integration of the neurovascular unit, where neurons, pericytes, astrocytes, microglial cells, and endothelial cells interact to form very tight junctions with the presence of polarized efflux pumps on the luminal surface.

The complexity of the model has led to the suggestion of simpler approaches, such as using only brain microvascular endothelial cells, for example [123]. Instead, other works have explored more sophisticated approaches that include endothelial cells in a matrix of collagen with astrocytes and pericytes [114]. After five days, the astrocytes and pericytes associated with the endothelial cell monolayer, and a functional barrier was formed that prevented the diffusion of large proteins such as bovine serum albumin but also proteins of lower molecular weight. Researchers have explored complex and specific configurations of the chips to accommodate other types of physiological barriers with the vasculature. Indeed, a sophisticated model design has been developed to simulate epithelium or alveoli, incorporating flexibility and mechanical stretch, two independent flow channels with the possibility of introducing air into one of them, and an intermediate interface space for cell culture in a biomimetic matrix [129].

#### 3.2.2. Angiogenesis

Angiogenesis or neoangiogenesis is one of the most studied functions using the new models. An interesting review of the last 10 years, about the microvascular chips created for the study of angiogenesis, vasculogenesis and lymphangiogenesis, summarized the main components required to produce a self-assembled vascular network on a chip, including the endothelial cell source, tissue-specific supporting cells, biomaterial scaffolds, biochemical cues, and biophysical forces [130]. The incorporation on the chips of biocompatible or biological ECM surrounding or in contact with the endothelial cell monolayer allows for the possibility of these cells to sprout, generate new vessels, and even create a perfusable functional capillary network. This process can be enhanced by the incorporation of fibroblasts or mesenchymal cells into the ECM or by gradient infusion of proangiogenic factors [102,104,114]. In addition to the cells in the matrix (simulating the tissue surrounding the vessels) and the proangiogenic factors, the new vessel-on-a-chip models allow for the study of the influence of factors such as interstitial flow, which seems to trigger the mechanosensing systems of endothelial cells to promote vasculogenesis [131]. Altogether, these results from angiogenic models are useful for research in several fields, including the study of various triggers of the process, neovessel formation in pathologies such as tumors or retinal diseases, or even the creation of a functional and physiological capillary network for incorporation into multi-organ-on-a-chip systems, as we will see below.

A system consisting of two flanking channels leaving a central space filled with collagen (or other biological matrix), in contact with the channels, allows for the creation of a gradient of angiogenic factor that diffuses from one channel to the other. Endothelial cells can grow to cover the lumen and react to the signal of the angiogenic factor, reproducing the dynamics of sprouting and filopodia projection observed in vivo during the development of blood vessel networks [113]. Morphogenesis can be studied in response to different factors or only to the interstitial flow using an appropriate microscopic analysis. Similar models have also been created [102,114]. In an attempt to translate the idea of these models to a multi-chip platform, a system called “Anchor-IMPACT” was created based on the gel injection on an anchor-based well adapted to the standard multi-well culture plates. This system allows for the formation of vasculogenesis induced by different cell lines of cancer and the study of different anti-angiogenic drugs in the setting, with the advantage of being compatible with automatic analytical equipment for culture plates [132].

If pericytes are included in the ECM, their recruitment and accommodation to the newly formed vessels can also be observed [104,114]. More specific systems have been developed to study vasculogenesis in the eyes, with the aim of exploring inward and outward angiogenesis in the eyes and serving as in vitro models for drug functional testing. These systems can also help to reduce the need for animal experimentation for this purpose [133].

#### 3.2.3. Capillary Perfusable Network

Neoangiogenesis should evolve to a functional capillary perfusable network. The microfluidic models have shown to allow this progression, providing a system where the molecular pathways of the process can be studied [113]. A faithful reproduction of the in vivo dynamics of sprouting and filopodia projection of endothelial cells for the development of blood vessel networks make the model relevant for this kind of mechanistic studies, and different combinations of cells in the ECM have been studied to induce the capillary network formation [134,135]. Apart from the development process of the capillary formation, the properties of these capillaries can be studied in the models. Barrier properties and cell-to-cell junction proteins in these microvessels can be analyzed, and the effects of different drugs or stimuli can be reproduced/studied [128]. Even the specific functionality of the new capillary network can be harnessed to interconnect reservoirs inside the same organ-on-a-chip, between different chips, part of the chips, or with the plastic connectors (inlets or outlets) of the microfluidic chip, as a model of anastomosis [106,136]. All these models try to resemble the self-promoted vasculogenesis in tissues. However, we should not forget that new advanced technologies could offer the possibility of recreating complicated microvascular 3D architecture with a high level of definition. This is the case for the combination of a two-photon polymerization laser technique for the high-definition bioprinting of microvascular networks directly on-chip [99]. Or, even more promising, we should consider that these high-definition microfabrication techniques can be combined with the self-induction microvascular network formation demonstrated in the previous models.

Capillary perfusion can be evaluated in vitro by measuring the ratio of fluorescence intensity of a substance in vascularized tissues compared to unvascularized tissues [137]. This also allows for the replication of changes in capillary perfusion in response to pathophysiological situations such as inflammation (which increases capillary permeability) or ischemia (which reduces vessel permeability and results in the failure of irrigation to a particular region) [138].

#### 3.2.4. Interaction with Circulating Cells

Finally, one of the main issues in validating the pharmacological relevance of microfluidic models is the ability to analyze cell behavior in a biomimetic environment, including cell movements and interactions between different cell types. In vessel-on-a-chip models, this issue is related to the interactions between circulating cells and the endothelium in vessel walls as well as the possible transmigration of these cells across the endothelial monolayer. Recapitulating in the vessel-on-a-chip models the dynamics of the interaction between the circulating cells, mainly leukocytes, and the endothelial cells in the walls of the vessels is a challenge that has been approached in some models [139]. The adhesion of circulating monocytes to a multi-chip platform of microvessels covered with human coronary artery endothelial cells has been tested in the presence of tobacco molecules with or without the stimulus of tumor necrosis factor-α [140].

In another model, the entire process of leukocyte adhesion, diapedesis, and extravasation over an endothelial cell monolayer seeded in a porous filter on a collagen matrix with a chemokine gradient was monitored using a real-time microscopy acquisition system integrated into the microfluidic system [103]. Murine neutrophils were introduced into the flow over a monolayer of brain endothelioma cells (bEnd.3) [103], and similar models have allowed for the observation of the response of microvessel models to stimulation with biological factors such as tumor necrosis factor-α [105]. Moreover, more complex models can be configured to include the analysis of blood constituents, such as red blood cells [141].

More aggressive models have been created to reproduce the interaction of endothelial cells and the extravasation of circulating tumor cells. Cells with different metastatic potential were introduced into endothelial capillary networks created in the fluidic chip to track their transmigration potency [114]. This model was able to reproduce steps of the metastasis process. Cancer cell extravasation is affected by two main factors: biochemical signals from specific organs and circulation patterns of the cardiovascular system. This obligates researchers to work with complex and perfusable physiomimetic vascular networks as well as to create organotypic microenvironments with specific endothelium [142]. Future steps could include the inclusion of a lymphatic system, for example.

Live-cell microscopy could complement the analysis of cells by tracking their positions within a defined time interval, which could provide a quantitative assessment of mitosis rate, apoptosis process, or cell motility [123,137]. Changes in any of these parameters are typically associated with a change in the morphology of endothelial cells [143,144]. Additionally, an increase in the rate of structural protein synthesis is associated with changes in cell structure and function [145]. However, the unique characteristics of certain microenvironments can result in differences in cell behavior. For instance, in brain vessels, the tight junctions between endothelial cells may make them resistant to elongation induced by shear stress [146].

### 3.3. Advantages of Vessel-on-a-Chip Systems with Respect to 2D Classical In Vitro Models

Based on the data and explanations presented above, we can review the main advantages of microfluidic 3D systems compared to classical 2D in vitro models. The two main characteristics of microfluidic systems are the incorporation of flow and the possibility of a 3D spatial configuration. Flow is present in all tissues at a wide range of rates, from the high velocities of blood in large vessels to the low flow rates in the interstitial space. This makes microfluidic models more realistic and important for functions that are governed or affected by mechano-sensing. Mechano-transduction mechanisms can be studied in these systems because different flow patterns can be created, and the different wall shear stress forces can be studied [120]. An excellent example of how the fine-tuning characteristics of interstitial flow can influence the response of endothelial cells is in angiogenesis. The mechanical environmental context defined by interstitial flow influences the initiation, outgrowth, and phenotype of angiogenic sprouts of endothelial cells [147]. This demonstrates that many parameters that could not be included in 2D in vitro models are crucial for understanding pathophysiological processes that contribute to the development of human diseases. Some of these parameters can now be considered in new microfluidic models. The combination of flow and ECM allows for the creation of physiological gradients of factors on this matrix to recreate biological signaling in tissues [103].

The response of cells to any stimulus can differ in a 3D environment compared to a classical 2D configuration, causing us to reconsider much of our previous knowledge about in vitro cell behavior. For example, Locatelli et al. [148] showed that the response of endothelial cells to high glucose depends, at least, on the tissue source of the cells and the flow regimen (i.e., shear stress intensity), and consequently, it differs from the response observed in the same cells cultured in 2D.

The incorporation of flow also allows for the introduction of different components, from molecules to cells, into the flowing solution. Thus, in addition to the mechanical flow stimulus, chemical or biochemical stimuli can be incorporated into the models in the same solution that exerts the mechanical forces [120]. On the other hand, the 3D spatial configuration is critical for many biological functions that are poorly mimicked in 2D classical models. Examples include BBB models, angiogenesis [102], vascular permeability [128], and cellular extravasation [114].

Combining all these factors in microfluidic chips allows for the study of physiological transport across functional biological barriers in a more realistic way than in 2D in vitro models. This has been implemented for biological functional units such as the BBB, alveolar–capillary interface, renal vascular–tubular unit, and placenta [149].

Importantly, the 3D biomimetic structure, the biological recreation of the microenvironment, and the presence of continuous flow seem to help maintain the original cellular phenotypes in vascular models [115].

In summary, the discussion so far highlights the potential of microfluidic chips in mimicking blood vessels in vitro. These chips can reproduce or predict vascular responses, specifically the endothelium’s response to stimuli like SARS-CoV-2 infection, in a highly controlled environment with sufficient quality. Vessel-on-a-chip models have demonstrated the ability to mimic endothelial dysfunction in terms of inflammatory response, and the interaction of circulating cells with the endothelium can also be analyzed in a physiological environment (Figure 2).

While there has not been enough time to develop microfluidic models that incorporate the analysis of the effects of SARS-CoV-2 or any of its proteins, the technology is readily available. These models could be used to investigate the molecular pathways involved in the response, specifically the ACE/ACE2 ratio, which appears to be at the core of the vascular reaction to COVID-19. By accurately reproducing the pathophysiological response, such models could shed light on possible changes in enzyme expression or activity as well as downstream signaling pathways leading to endothelial dysfunction. Ultimately, these insights could identify appropriate therapeutic targets for combating the virus (Figure 1). The next technological step would be the “personalization” of the in vitro models and adapting them to “point-of-care” study models, as discussed below.

The flow of organs-on-a-chip offer important improvements compared to 2D in vitro models; however, they also face some challenges that have been grouped as follows [120]: (1) establishing a mechanical mapping of different cell types responses to flow to understand the correlation between flow velocity, temporal pattern, and cell activity (characterization of mechano-adaptation); (2) multiplexing; (3) integrating novel complementary modules to account for various microenvironmental cues; (4) developing commercially available, miniaturized, ready-to-use pumping modules with the capacity to produce a wide range of flow rates and patterns; and (5) improving access to cells or measurements in the chip.

The last challenge is related to the characterization techniques for measuring inside the chip. The limited access to the intricate 3D structure requires the use of high-resolution microscopes that can accommodate a wide range of working distances on their lens. So, the ideal chip should allow for the observation by phase-contrast, fluorescent tagging, or confocal microscopy, of parameters such as cell’s deformation, surface area, orientation angle, adhesion, migration, and proliferation. Other methods such as immunostaining or nucleic acid quantification sacrifice the cells, so they do not provide real-time data. Complementary methods are used to prove real-time cell activity: biomolecules uptake and secretion, live cell labeling, trans-endothelial electrical resistance assay (TEER), cell transfection, and cell permeability assays [120].

A tentative approach to solve this issue has been the creation of a multiscale cytometry analytic platform with a combination of microfluidics, 3D culture of spheroids and 2D culture of endothelial cells [150]. This platform allows the statistical analysis of hundreds of spheroids and thousands of individual cells on those spheroids.

### 3.4. “Personalization” of Endothelial Models

The evolution and increasing accessibility of gene editing techniques and induced pluripotent stem cells (iPSCs) have enabled the development of personalized in vitro models. Cells with a genome profile that resembles a rare pathology or cells from a particular patient can now be generated using these techniques. When combined with microfluidic systems/organ-on-a-chip, these techniques hold great promise for: (1) disease modeling, avoiding the need for variable in vivo experimentation in early drug development; (2) high-throughput compound screening; (3) preclinical studies, provided that standardized assays are developed; and (4) patient-specific studies for risk stratification, efficacy, or toxicity [151]. However, standardization of the models and assays employed in the organ-on-a-chip industry remains a major challenge [152]. Nevertheless, industry stakeholders have accepted this challenge and are working together to harmonize the field [153].

The second big challenge in standardization is the robustness of the differentiation protocols from iPSC to obtain the vascular cell types and the phenotypic characterization of these cell types and their specific functions in the space of the organ-on-a-chip. A differentiation protocol for endothelial cells has been demonstrated that maintains vasculogenic properties and resembles the sensitivity to anti-angiogenic drugs [154]. Additionally, new methods have been reported for differentiating endothelial cells and pericytes from keratinocyte-derived iPSCs obtained from patients, which could aid in obtaining personalized vascular models for any patient [155]. However, it is important to consider incorporating organ-specific endothelial cells into the assays, as tissue specificity appears to be a factor, at least in the endothelial extravasation response [156].

In any case, vascular models employing endothelial cells differentiated from iPSCs have been developed in combination with pericytes [157]. Furthermore, a vessel-on-a-chip using iPSC-derived endothelial cells and vascular smooth muscle cells was achieved, in which functional responses in smooth muscle cells were demonstrated, and an automated analysis of microvascular network morphology and intracellular calcium release could be performed [158].

The participation of endothelial cells in complex in vitro models of heart-on-a-chip, in combination with cardiomyocytes, has also been demonstrated using cells derived from iPSCs. In these settings, these cell types retained their phenotypes and functions, confirming their utility for personalized medicine [119]. Some of these models even allow for the simultaneous measurement of intracellular calcium movements and electric pacing, making them particularly powerful tools for studying cardiac function [159]. A particularly important application of these models in personalized medicine is their utility to test the cardiotoxicity of anticancer drugs [98,159].

Therefore, the combination of genetic modified cells and patient-derived cells in the microfluidic platforms has allowed the study of the molecular mechanisms underlying the cardiovascular diseases that were not possible with the standard in vitro or animal models of the diseases [160].

### 3.5. Tools for “Point-of-Care” Solutions, Diagnosis or Follow-Up

As discussed so far, vessel-on-a-chip models have been developed for studying vascular functions in a biomimetic environment with flow. If these models can be personalized based on individual patient response or the specific behavior of rare diseases, the next question is whether they could be used as point-of-care solutions for more precise diagnosis, risk stratification, or early detection of emerging diseases such as COVID-19.

The vessels-on-a-chip developed for testing cardiotoxicity of antineoplastic drugs [98,159] opened the door for drug screening against a particular cardiovascular problem. In fact, the idea of point-of-care solutions for assessing the risks associated to atherosclerosis plaque reaction in patients with diabetes or dyslipidemia was proposed using a vascular model of stenosis seeded with endothelial cells and where the patients’ blood samples were tested for their particular responses [161].

One crucial point for the purpose of point-of-care is the miniaturization of the assays in microfluidic chips and the possibility to include it in a multi-chip platform to guarantee the volume of workflow of the clinical care process. Some attempts have been made in this context in models for vasculogenesis [132] or for monocytes adhesion to endothelial cells [140]. It may also be useful to consider incorporating microfluidic chips into conventional multi-well culture plates, such as the ‘insert-chip’ described by Rauti et al. [162], in order to streamline the workflow of clinical care processes.

Interestingly, the evolution of this idea of point-of-care for vascular toxicity testing has arisen from studies on air pollutants. The effects of airborne nanoscale particles on microvascular networks generated in 3D microfluidic devices have been tested to evaluate the reaction of endothelial cells [163]. So, the model is useful for testing vascular toxicity after exposure to air pollutants [164]. An improved version of the system includes the interaction of endothelial cells with alveolar epithelial cells [165]. All this progress is of particular interest, given that the analysis of air pollutants could include respiratory viruses that enter the body through the alveoli, such as SARS-CoV-2.

However, for these chips to be adopted for the purpose of point-of-care solutions, miniaturization and multi-chip development would be necessary. In this sense, very recently, a 64-chip microfluidic plate-based platform has been adapted for a vascularized lower respiratory tract model of alveoli [166]. Another multi-chip vascular model has been developed for a plate platform to study monocyte adhesion to endothelial cells [140].

The real proof of concept for the utility of vessel-on-a-chip in COVID-19 demanding situations has been reported recently. A vessel-on-a-chip with innate immunity response was used to study SARS-CoV-2-induced inflammation in vascular tissue. The model measured the endothelial barrier function in response to the interaction of endothelial cells with the virus and tested the effects on endothelial cells’ response to the introduction of peripheral blood mononuclear cells after this interaction. The results of the study support the idea of the need for an inflammatory attenuation treatment to protect vascular tissue integrity in the case of SARS-CoV-2 infection [167]. Another study in a 3D microfluidic model of the human BBB revealed that the SARS-CoV-2 spike proteins trigger a pro-inflammatory response in brain endothelial cells that may contribute to an altered state of BBB function [168]. All of these approaches aim to simplify models in order to miniaturize them and include them in multi-chip platforms that could be adopted as point-of-care solutions. However, more complex solutions that aim for a complete understanding of the problem have been suggested, at least in other fields. The interaction with the vasculature requires the use of complex and perfusable physiomimetic vascular networks and the creation of organotypic microenvironments and specific endothelia. The next steps should include drainage systems, such as the lymphatic system, metabolizing units that resemble the liver, and a mimetic vascular system that connects the multi-organ system [142]. This vision may lose the point-of-care utility, but it would provide a global point of view that helps predict the behavior of diseases, which is of particular importance in the case of new diseases.

## 4. Concluding Remarks

SARS-CoV-2 interacts with the RAS because it can disrupt the homeostatic balance between ACE and ACE2 in this system. The role of RAS in diseases such as heart failure, myocardial infarction, hypertension, kidney diseases, acute lung injury, or diabetes mellitus has been demonstrated for decades. Many of these cardiorenal and lung diseases have been reported as significant comorbidities that increase the complications, severity, and outcomes of COVID-19.

The interaction of SARS-CoV-2 with ACE2, inducing endothelial dysfunction, could be the basis of COVID-19 complications and many of its comorbidities. Therefore, one of the lessons from the pandemic situation is that more investigation into the pathophysiology of COVID-19 in endothelial cells is needed to identify new therapeutic targets and drugs that can help reduce the incidence of COVID-19 complications and their severe consequences. The personalization of follow-up strategies for post-COVID-19 patients (eventually all of us) should include sophisticated tools to analyze endothelial function, which allows individualization of the assay according to each person’s conditions.

Vessel-on-a-chip models for COVID-19 would allow the study of the disease’s pathophysiology in endothelial cells by analyzing the interaction between SARS-CoV-2, endothelial function, leukocyte recruitment, and platelet activation. Personalization could be achieved in these models through the use of iPSC, patient-specific characteristics, or genetically modified cells. The implementation of point-of-care characteristics could ensure mass testing under standardized protocols. These chips could also be used for personalized follow-up of the disease or its sequelae (long-COVID) as well as for research into new drugs against COVID-19 or for testing the effects of common cardiovascular drugs. All of this is now possible due to the development of microfabrication techniques for biomimetic miniaturization, which allow for pharmacological relevance in the important functions of the endothelium.

## Figures and Tables

**Figure 1 cells-12-01297-f001:**
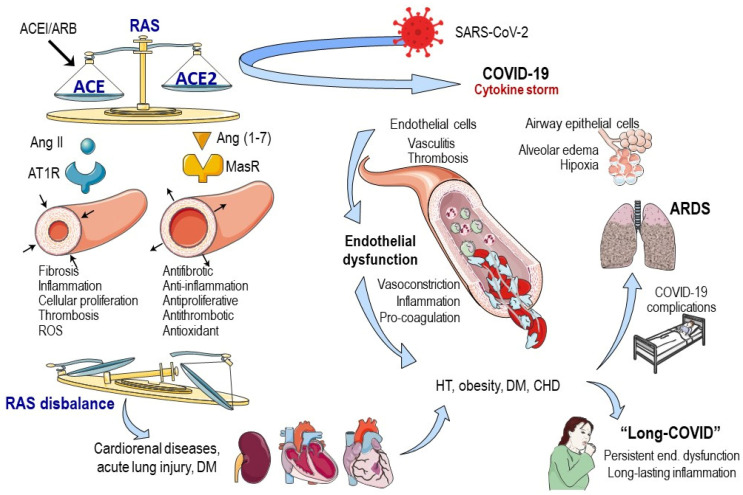
Participation of vascular endothelium in COVID−19 pathophysiology. Homeostatic behavior of renin–angiotensin system depends on the balance of ACE/ACE2, and it has a role in several diseases such as heart failure, myocardial infarction, hypertension, kidney diseases, acute lung injury or diabetes mellitus. The interaction of SARS-CoV-2 with ACE2 can alter this balance, inducing endothelial dysfunction that could be on the basis of COVID-19 complications and many of its comorbidities. The situation of long COVID (COVID−19 symptoms persistence without virus infection), characterized by persistent endothelial dysfunction and long-lasting inflammation, could be related with COVID−19 severity and/or with ACE2 expression in tissues such as the endothelium. Abbreviations: ACE(2): angiotensin converting enzyme (2); ACEI: angiotensin converting enzyme inhibitors; Ang(1–7): angiotensin (1–7) peptide; Ang II: angiotensin II; ARB: angiotensin receptor blockers; ARDS: acute respiratory distress syndrome; AT1R: angiotensin type 1 receptor; CHD: coronary heart disease; COVID-19: coronavirus disease; DM: diabetes mellitus; HT: hypertension; MasR: mas receptor; RAS: renin–angiotensin system; ROS: reactive oxygen species; SARS-CoV-2: severe acute respiratory syndrome coronavirus 2.

**Figure 2 cells-12-01297-f002:**
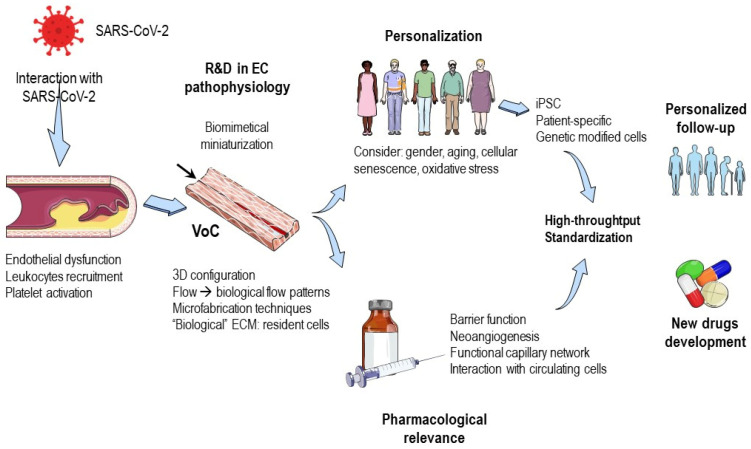
Vessel-on-a-chip (VoC) opportunities in COVID-19 research. VoC models for COVID-19 would allow the study of the pathophysiology of the disease in endothelial cells, analyzing the interaction of SARS-CoV-2 with endothelial function, leukocytes recruitment and platelet activation. In these models, “personalization” could be implemented through induced pluripotent stem cells, the implementation of patient-specific characteristics or genetic modified cells. Point-of-care characteristics should be implemented in the models to guarantee massive testing under standardized protocols. These characteristics could be incorporated for the personalized follow-up of the disease or its sequalae (long COVID) and for the research of new drugs against COVID-19 or to test the effects of common cardiovascular drugs. All of this is now possible thanks to the development of the microfabrication techniques for biomimetical miniaturization, which allow the needed pharmacological relevance in important functions of the endothelium. Abbreviations: EC: endothelial cells, ECM: extracellular matrix, iPSC: induced pluripotent stem cells, R&D: research and development, SARS-CoV-2: severe acute respiratory syndrome coronavirus 2, VoC: vessel-on-a-chip.

## Data Availability

No new data were created or analyzed in this study. Data sharing is not applicable to this article.

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
