# Peer review of "Vessel-on-a-Chip: A Powerful Tool for Investigating Endothelial COVID-19 Fingerprints"

_cells, 2023, doi:10.3390/cells12091297_

Round 1
Reviewer 1 Report
Shevchuk and colleagues comprehensively reviewed vascular consequences associated with SARS-CoV-2 infection and examine the potential role of vessel-on-a-chip models in COVID-19 research. Application of suitable models could potentially allow important analysis of the interaction between SARS-CoV-2 and vascular endothelial cells, including changes in endothelial function, ACE/ACE2 balance, cytokine/chemokine response, leukocyte recruitment, microvascular thrombosis, etc. The alteration of ACE/ACE2 balance in response to SARS-CoV-2 infection was extensively discussed in this review, however, this important topic was not included when analysing the utility of microfluidic in vitro models. Some reflection on how microfluidic in vitro models could be applied to this crucial topic will be beneficial for COVID-19 research.
Author Response
Thank you very much for your wise comments. As you suggest, we have improved the manuscript including a brief discussion of how the microfluidic in-vitro models could be used to study the ACE/ACE2 ratio in COVID-19 (lines 653-669).
Reviewer 2 Report
The authors have comprehensively searched the databases to provide current knowledge on the main vascular sequelae of COVID-19 including post-acute COVID-19 and post COVID-19 (long COVID-19) effects and available 3D tools for performing mechanistic studies.
Mounting evidence suggests that SARS-CoV-2 infection leads to multiple instances of endothelial dysfunction, including reduced nitric oxide (NO) bioavailability, oxidative stress, endothelial injury, glycocalyx/barrier disruption, hyperpermeability, inflammation/leukocyte adhesion, senescence, endothelial-to-mesenchymal transition, hypercoagulability, thrombosis and many others. Thus, COVID-19 is deemed as a (micro)vascular and endothelial disease. However, the pathophysiology of acute and post-acute manifestations of COVID-19 (long COVID-19) is understudied. The authors have made an excellent effort covering several aspects of COVID -19 related dysfunction and identifying the potential benefits of emerging new microfluidics, vessel on-a chip technology to increase our understanding of the underlying mechanisms and integrate clinical findings. The authors have provided their thoughts and recommendations on how “Personalization” could be implemented through clinical finding in these models, that could allow personalized follow up of post COVID-19 effects and new drug development taking into considerations the involved challenges. The review article is well articulated and will benefit the researchers in the field. Very minor correction is required as follows:
Please replace "thanks" with "due to" in line#793
Author Response
Thank you very much for your wise comments. As you suggest, we have replaced “thanks to” by “due to” (line 809 in the new version).